# Android in the Wild: A Large-Scale Dataset for Android Device Control

**Christopher Rawles**[*]
Google Research

**Alice Li**[*]
Google Research

**Daniel Rodriguez**
Google Research

**Oriana Riva**
Google Research

**Timothy Lillicrap**
Google DeepMind

## Abstract

There is a growing interest in device-control systems that can interpret human natural language instructions and execute them on a digital device by directly controlling its user interface. We present a dataset for device-control research, Android in the Wild (AITW), which is orders of magnitude larger than current datasets. The dataset contains human demonstrations of device interactions, including the screens and actions, and corresponding natural language instructions. It consists of 715k episodes spanning 30k unique instructions, four versions of Android (v10–13), and eight device types (Pixel 2 XL to Pixel 6) with varying screen resolutions. It contains multi-step tasks that require semantic understanding of language and visual context. This dataset poses a new challenge: actions available through the user interface must be inferred from their visual appearance, and, instead of simple UI element-based actions, the action space consists of precise gestures (e.g., horizontal scrolls to operate carousel widgets). We organize our dataset to encourage robustness analysis of device-control systems, i.e., how well a system performs in the presence of new task descriptions, new applications, or new platform versions. We develop two agents and report performance across the dataset. The dataset is available at `https://github.com/google-research/google-research/tree/master/android_in_the_wild`.

## 1 Introduction

Users complete tasks on mobile devices via a sequence of touches and gestures on the screen. Tasks can often be succinctly described using natural language commands, and, in many situations, it is valuable to be able to speak or type commands rather than interacting directly with the device. This has important implications for users who are unable to physically operate a device due to a physical (e.g., visual or motor disabilities) or situational (e.g., driving, cooking, etc.) impairment. It is therefore beneficial to build device-control systems that can interpret natural language instructions and execute them on a device without any manual intervention.

Instead of using application-specific APIs, which are not generally available for any given application or function, these systems directly manipulate user interface (UI) elements on a screen, exactly as a human does [1, 28, 29, 35, 21]. Hence, to work correctly, it is essential for such systems to understand the screen, which usually means detecting position and inferring semantics of its UI elements. Device-control systems must also be able to map high-level commands to execution plans that can be carried out on the device. For example, understanding that the command "*open my recent email with Jane*" involves opening an email app, potentially tapping the search icon, typing "Jane", etc. Further, to be useful, they must be able to generalize across a variety of task instructions and UIs.

---

[*]Equal contribution. Contact: `crawles@google.com` and `lialice@google.com`

37th Conference on Neural Information Processing Systems (NeurIPS 2023) Track on Datasets and Benchmarks.

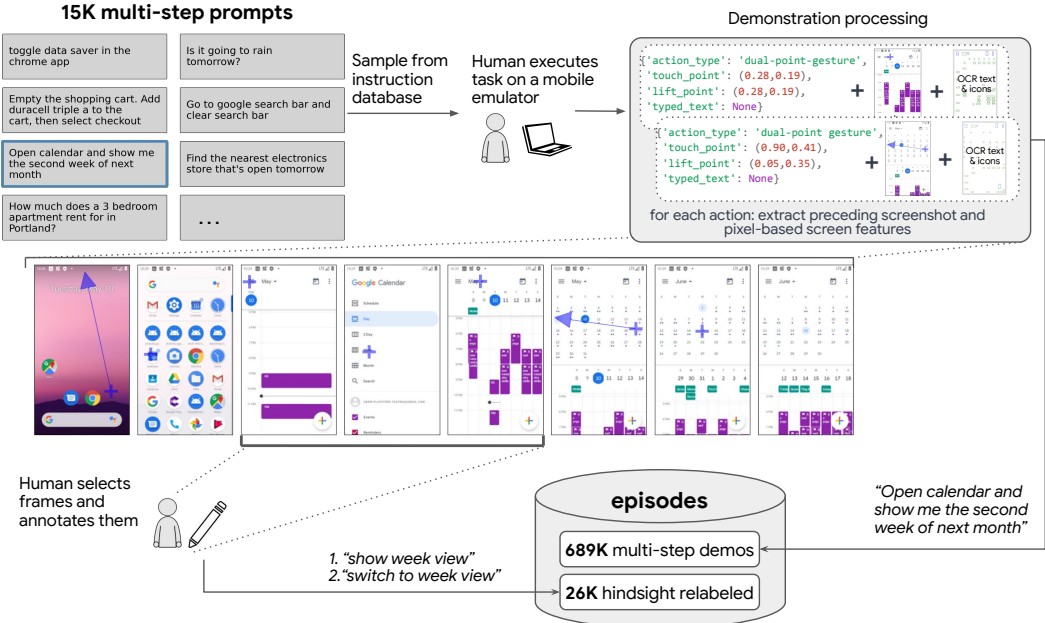

Figure 1: **AITW data pipeline**. Raters are given a randomly selected instruction. The raters execute the task by interacting with the device in a natural way. We capture precise gestures in addition to typing and the home and back button interactions (we plot swipes with the arrow pointing where the finger moves to). Hindsight relabeling of high-level episodes is used to generate single-step tasks.

The rapid development of general-purpose large foundation models (LLMs) [8, 6, 13] makes device-control systems more viable. Yet, there is a lack of datasets for training, fine-tuning, and evaluating these systems. Existing datasets [28, 9, 42, 37, 4] are limited in terms of number of human demonstrations and the diversity of task instructions, and they are platform specific (either Android or web). They also assume a tree-based representation of an application UI can be derived from platform-specific UI metadata (e.g., the View Hierarchy for Android and the DOM tree for the web). This assumption simplifies the problem, but limits the resulting systems to work in environments where high-quality UI metadata is available[2]. Finally, some popular datasets (e.g., MiniWoB++ dataset [29] and UIBert [4]) assume task instructions are specified as step-by-step commands referring to specific UI elements appearing on the screen (*"Click the button in the dialog box labeled Cancel"*), while users may use short commands that describe high-level goals (e.g., *"turn on airplane mode"*) or pose questions (e.g., *"Is it going to rain tomorrow?"*)

To drive research in this field, we release AITW (Figure 1), an Android device-control dataset which is orders of magnitude larger than existing datasets. It consists of 715k episodes spanning 30k unique task instructions collected across hundreds of Android apps and websites. Each episode consists of a goal instruction provided in natural language and a sequence of observation-action pairs describing the execution of the task. Observations consist of screenshots of the application UI. Gesture actions are represented as taps and drags at arbitrary <x,y> coordinates in the screen. Agents trained on this dataset can be evaluated using AndroidEnv [40], an open-source platform for developing and testing Android agents with the Android Emulator[3].

A key feature of our dataset is the diversity of task instructions and execution paths we collected, aimed to emulate real-world scenarios. We used multiple sources to collect high-level goal instruc-

---

[2]Since most users do not use UI metadata for interactions it tends to be poor quality or missing altogether. On Android, only applications registered as Accessibility tools can access the View Hierarchy [45]. On Windows, in many cases (e.g., Electron apps like Teams), UI trees are not easily accessible. Moreover, screen representations derived from UI metadata can be incomplete. On Android, WebViews and Canvas are not captured in the View Hierarchy, and many websites render directly to a Canvas, which does not contain any tree structure.

[3]https://developer.android.com/studio/run/emulator

| Dataset | Platform | # Human demos | # Apps or websites | # Task steps | Observation format | Screen features | Real | High-level instruction |
|---|---|---|---|---|---|---|---|---|
| RicoSCA [28] | Android (apps) | 0 | n/a | 1.0 | VH, screen | x | x | x |
| UIBert [4] | Android (apps) | 16,660 | n/a | 1.0 | VH, screen | x | ✓ | x |
| MiniWoB++ [37, 29] | synthetic web | 17,971 | 100 | 2.3 | DOM, screen | x | x | x |
| PixelHelp [28] | Android (apps) | 187 | 4 | 4.2 | VH, screen | x | ✓ | ✓ |
| UGIF [42] | Android (apps) | 523 | 12 | 5.3 | VH, screen | x | ✓ | ✓ |
| Mind2Web [14] | web | 2,350 | 137 | 7.3 | DOM, screen | x | ✓ | ✓ |
| MoTIF [9] | Android (apps) | 4,707 | 125 | 4.5 | VH, screen | x | ✓ | ✓ |
| **AITW** | Android (apps+web) | 715,142 | 357+ | 6.5 | screen | ✓ | ✓ | ✓ |

Table 1: Comparison of AITW to existing datasets. We consider platform, format of screen observations, presence of synthetic UIs or synthetic instructions ("Real"), and whether instructions are expressed as goals (high-level). For size comparison, we report the number of human demonstrations, apps/websites, and average task steps. AITW collects observations as screenshots and includes screen features (OCR and icon labels), which can be used to augment them.

tions: humans (both crowdsourced raters and us authors), LLM-generated prompts, and technical documentation (as in PixelHelp [28]). During crowdsourcing, raters were asked to both demonstrate full tasks and annotate sequences of screenshots (hindsight language relabeling [31, 32]), which allowed us to collect both multi-step and single-step task trajectories. We made the execution paths more varied by randomizing the application state, which forced the raters to demonstrate how to navigate to the relevant screens. Finally, we collected demonstrations on four versions of Android (v10–13) and eight device types (Pixel 2 XL to Pixel 6) with varying screen resolutions.

Device-control systems need to work on rapidly evolving software platforms, so an important metric for their success is generalizability to new tasks and applications. We organize our dataset to enable analysis of how trained systems perform in the presence of previously-seen tasks and applications, but also in the presence of new task descriptions, new Android versions, and new applications. Due to the lack of off-the-shelf pixel-based device-control models, to establish new state-of-the-art results on this dataset, we implement two agents: one trained from scratch using behavioural cloning (BC) and a second based on a pre-trained LLM.

We make the following contributions: *(i)* we collect and release a dataset for device-control research, AITW, which is larger and more varied than existing datasets; *(ii)* we report performance for two models, which can serve as baselines for future work and *(iii)* we show how to use the dataset to conduct a generalization analysis.

## 2 Related work

### 2.1 Device-control datasets

Table 1 provides a comparison of device-control datasets. Some datasets (top part of the table) target the problem of grounding referring expressions to UI elements on a screen. Every data instance in these datasets includes a screen, a low-level command (e.g., *"click the menu button at the top left"*), and a UI element corresponding to the command. In the RicoSCA dataset [28], commands are synthetically generated, while in MiniWoB++ [37, 29] sequences of low-level UI commands describe multi-step tasks (e.g., *"find and click on the center of the circle, then press submit"*).

A second group of datasets contain instructions expressed as task goals. Each episode in these datasets is a sequence of action-observation pairs. The observations include screenshots and tree-based representations: View Hierarchy (VH) for Android and Document Object Model (DOM) for web-based applications. For instance, the PixelHelp dataset [28] comprises 187 high-level task goals and step-by-step instructions sourced from Pixel Phone Help pages. The UGIF dataset [42] contains similar queries but extends to multiple languages. The largest dataset to date is MoTIF [9], which consists of 4.7k task demonstrations[4] with an average number of 6.5 steps and 276 unique task instructions. AITW is two orders of magnitude larger than MoTIF. In total, AITW consists of 715,142 episodes, spanning 30,378 unique prompts, with a small subset of the prompts inspired by

---

[4]This represents the number of "feasible" tasks. We do not consider tasks without a valid demonstration.

| Name | Task type | Description | Episodes | Screens | Prompts |
|---|---|---|---|---|---|
| GOOGLEAPPS | | Tasks with Google apps (Gmail, Photos, Settings, etc.) | 625,542 | 4,903,601 | 306 |
| INSTALL | Multi-step | App installation and login tasks | 25,760 | 250,058 | 688 |
| WEBSHOPPING | | Web shopping tasks | 28,061 | 365,253 | 13,473 |
| GENERAL | | Misc web/app tasks and Q&A | 9,476 | 85,413 | 545 |
| SINGLE | Single-step | Mostly shopping tasks from WEBSHOPPING | 26,303 | 85,668 | 15,366 |
| **Total** | | | 715,142 | 5,689,993 | 30,378 |

Table 2: Composition of the AITW dataset.

the PixelHelp dataset. Observations are represented by screenshots along with pixel-based screen features.

## 2.2 UI representation and automation models

Research on device control is mainly focused on two problems: understanding UIs and automating tasks. Existing work on the first problem utilizes self-supervised [21, 4, 5, 27] and supervised methods [10, 30, 11, 46] to train UI understanding models. In some cases, these models are fine-tuned for simple grounding tasks (e.g., referring expression component retrieval [4]), along with widget captioning or question answering tasks [27, 4].

For task automation, Li et al. [28] decompose the problem in two stages: an action phrase-extraction stage to transform step-by-step instructions into actionable phrases, and a grounding stage that executes these instructions. Venkatesh et al. [42] utilize an LLM to parse the instruction before executing "macros" (e.g., tap(), toggle()) during the grounding phase. AppBuddy [38] train an RL agent to interact with on-screen UI elements to achieve tasks. LLMs can also understand and operate UI screens [43]. On the web front, previous studies have developed RL [20, 25, 29, 18], behavioral cloning [24], and LLM-based models [19, 26, 17]. These approaches utilize Document Object Model (DOM) inputs and often evaluate results on a simulated environment, MiniWob++ [37, 29]. Finally, LLMs have shown impressive results leveraging APIs, when they are available, for performing higher-level tasks [36, 33, 34].

## 3 Android in the Wild (AITW)

Table 2 shows the composition of AITW in terms of category and type of tasks. Overall, AITW consists of four multi-step datasets, GOOGLEAPPS, INSTALL, WEBSHOPPING, and GENERAL, along with a single-step dataset SINGLE.

The dataset is collected in a two-stage pipeline shown in Figure 1. First, we ask the raters to perform end-to-end tasks on emulators. Then the raters apply hindsight language relabeling [31, 32] to the trajectories that were collected in the first stage. We ask the raters to identify and label simple action sequences. We refer to these as single-step tasks.

Our recording system uses AndroidEnv [40] with the Android Emulator. The environment supports 3 action types {TOUCH, LIFT, REPEAT} with an (x,y) tuple indicating the on-screen position of the action. We record the TOUCH and LIFT actions. In response to an action, the environment returns an RGB screenshot, along with additional metadata such as the opened application. Raters interact with the emulated device using a mouse and keyboard on a desktop computer. Click events are logged as touch events. We provide dedicated buttons for Home, Back and Enter actions along with a field for entering text. We encourage the raters to use the dedicated buttons when necessary, however we require them to use a dedicated input text field for typing; we do not allow them to use the on-screen keyboard. We also ask the raters to indicate when they have completed the task or if they deem the task to be impossible to complete by pressing a button on our data collection UI.

The system captures the raw observations and actions at 10Hz. Mouse presses and releases are recorded as TOUCH and LIFT, respectively. For touch events, we log the start and end position of the virtual finger's gesture, which we call a "dual-point" gesture. A scroll is represented by a start and end position, and a tap is a special case where the start and end are approximately equal (<= 0.04 Euclidean distance away). Figure 1 contains an example of a tap and horizontal scroll gesture using

this formulation. We found the dual-point gesture abstraction to be a good trade-off between data compression and precision, allowing us to represent arbitrary drags that are needed to operate widgets, including scrolling through a menu and operating carousel widgets. After identifying dual-point gestures, we drop LIFT actions. Button presses and type events are logged as additional actions types. For type events, we log the typed text.

In summary, AITW's actions are described by four fields: *type*, *touch_point*, *lift_point* (only for gesture actions), and *typed_text* (only for typing actions). The type field can be one of the following: *dual-point gesture*, *type*, *go_back*, *go_home*, *enter*, *task_complete*, or *task_impossible*.

We post-process RGB screenshots to map them to a set of detected UI elements. Each element has a bounding box and either OCR-detected text or an icon class label (one of the possible 96 icon types detected using IconNet [39]). The OCR outputs describe most of the text on the screen, although certain characters can be misidentified and text blocks are not always grouped as desired. Although this screen representation inferred from pixels is noisy and not as comprehensive as that obtained from UI metadata, we provide these features for convenience and expect developers will replace them with more powerful screen understanding models. We use these features for training and evaluating our models.

### 3.1 Multi-step task trajectories

We first create high-level task instructions from various sources: (1) the authors, (2) a subset of PixelHelp [28] instructions that were deemed achievable, and (3) an LLM prompted to generate instructions. Next, we randomly assign instructions to raters and they follow them to complete tasks. Every task requires multiple steps to be performed. For example, the task "*show my schedule for next week in Google Calendar*" could correspond to the following steps: 1) opening Google calendar, 2) selecting "week view", and 3) opening next week. For each episode, we reset the environment to a random starting screen.

We ask the raters to interact with the device in a natural way, to avoid clicking on anything unrelated to the task, and to avoid unnecessary scrolling. To help guide the raters we prompt them with the following "*Imagine a friend is asking you to perform the task on their phone...*" The raters end a task with a special "status" action: either *task_complete* or *task_impossible*. A task is deemed impossible when an invalid or unavailable instruction is given, e.g., "*turn on flashlight*" on an emulator or "*show my starred emails*" when the Internet is not available. For instructions that result in verification rather than a state change (e.g., if the prompt is "*Turn wifi off*" and WiFi is found to be already off), we ask the raters to mark the task as successful.

### 3.2 Hindsight language relabeling

Single-step task demonstrations cannot be collected in the usual way of giving raters instructions and asking them to solve end-to-end tasks, since they require the relevant preceding steps to be executed. For example, in the task we described above, we cannot ask the raters to demonstrate "*go to next week*" unless they are already in the week view of the calendar app. Rather than asking raters to manually perform the single steps, we utilize event-selectable hindsight language relabeling [31, 32] to label previously collected trajectories.

To collect single-step demonstrations, we provide the raters observation-action sequences of multi-step task trajectories and ask them to identify and annotate shorter sequences (around two to five frames). We instruct them to label single steps, e.g., "add item to cart", "show the settings", "show me my bookmarks". We ask that they label at least K subsequences (K >= 3 in our case) per video.

We instruct the raters to avoid the following words: "click", "select", "tap", "touch" or "scroll down/up/left/right", since these can be easily synthetically created, and instead ask them to write descriptive phrases that describe the result of the action (e.g., instead of "tap airplane mode", write the label "disable airplane mode").

### 3.3 Dataset summary

With reference to Table 2, we describe the 5 sub-categories of AITW.

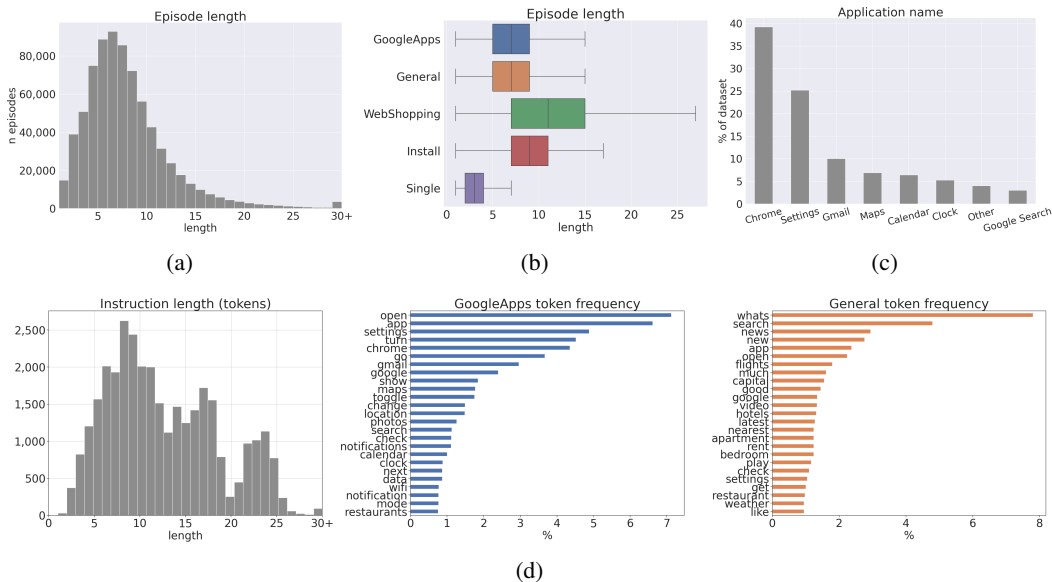

Figure 2: Statistics for AITW. a) Episode length distribution. b) Episode length distribution by dataset group. c) Frequency of Android apps in the dataset. d) Token analysis including distribution of instruction length and token frequency for GOOGLEAPPS and GENERAL.

**GOOGLEAPPS** contains high-level tasks with some overlap from PixelHelp [28] which involve various Google applications such as Gmail, Calendar, Photos, Settings, etc.

**INSTALL** contains high-level tasks related to installing and uninstalling apps, app login, and app login support (e.g., *"forgot password"*) for 88 different apps available on the Google Play store.

**WEBSHOPPING** contains tasks related to shopping on E-commerce websites. Example tasks include searching for an item, adding an item to the cart, viewing the shopping cart, etc.

**GENERAL** contains miscellaneous tasks (e.g., *"play the new Taylor Swift video on YouTube"*), mostly centered around question and answering (Q & A) (e.g., *"How much does a 2 bedroom apartment rent cost in San Francisco?"*) and interacting with third-party apps and websites.

**SINGLE** contains single-step tasks manually annotated using hindsight relabeling, mostly from WEBSHOPPING (e.g., *"Close the pop-up then add first item to cart"*,*"clear items from cart"*). It also contains a smaller amount of episodes (560) from a variety of Google apps and third-party websites.

In Figure 2, we report statistics about AITW. The episode length distribution (Figure 2a), measured as number of steps required to complete the task, shows that tasks are of moderate length (between 2 and 16 steps for the 5th to 95th percentile, respectively) and that WEBSHOPPING tasks are generally the longest (Figure 2b). Chrome and Google apps are the most commonly used apps (Figure 2c). Overall, the dataset spans 159 Android apps and 198+ websites.[5]

(Figure 2d) shows summary statistics of the instructions. Instructions lengths fall between 4 and 24 for the 5th to 95th percentile, respectively, and are *not* overloaded with technical terms such as "click", "tap", "menu", "button", etc. which is generally the case for low-level UI commands provided in existing datasets [37, 4].

## 4 Experimental setup

With the ultimate goal of building automation systems that can generalize to new scenarios, we use a standard test split and also design four experimental setups to evaluate Out-of-Distribution (OOD) generalization.

---

[5]This number is a conservative estimate computed using heuristics.

Table 3: Examples of subject templates.

| Instruction | Subject template | Split |
|---|---|---|
| open app grab and go to login screen | open {subject1} and | train |
| open walmart and go to shopping cart | go to {subject2} | train |
| search newegg.com on google | search {subject1} | val |
| search usb-c to usb-a on ebay | on {subject2} | val |
| add jbl flip 4 to the cart on bestbuy | add {subject1} to the | test |
| add acer nitro to the cart on target | cart on {subject2} | test |

**Standard.** We randomly split each dataset (the four multi-step datasets and SINGLE) episode wise into a training, validation, and test set (80/10/10%). Because the datasets different sizes, we evaluate each of them separately, then take the average score across them; we do the same for OOD setups.

**Unseen Android version.** To evaluate a system's performance on an unseen Android version — which contains unseen graphical components and execution flows — we partition our data as follows: We put episodes collected on Android versions 10, 11, and 12 into the training and validation sets, maintaining a 90/10% split respectively. Then, we create a separate test set comprising entirely of episodes captured on Android version 13 devices.

**Unseen subject and unseen verb.** This setup is aimed at evaluating generalization to unseen instructions. Due to the large number of prompts in AITW, it is infeasible to manually group similar tasks together. Simply splitting based on exact match of the raw instructions would be the most straightforward way to automatically assign splits. However, similar instructions with minor changes in language would potentially be seen in both training and testing.

To better differentiate the training and test sets, we develop instruction templates by masking out either verb or subject phrases (examples provided in Table 3). By splitting data based on these templates, we can assess a system's ability to generalize to unseen language patterns, and occasionally to entirely new tasks. For instance, all instructions following the template *add {subject1} to the cart on {subject2}"* are grouped together, ensuring they are not represented in both training and testing sets. Similarly, verb-based templates such as *open the shopping cart"* and "*view the shopping cart"* would be assigned to the same split.

We extract the templates for each instruction, by prompting a few-shot LLM [13]. In total, we extract 6,111 subject templates and 22,122 verb templates. For both types, we randomly assign each template to a train, validation or test split (with 80/10/10%). Then for each episode, we determine its template based on its instruction, and map the episode to a split.

**Unseen domain.** This split is designed to test an agent's ability to generalize to unseen apps and websites, which we refer to as *domains*. For WEBSHOPPING and GENERAL, we perform the split based on the web domain, as inferred from the instructions. For INSTALL tasks, we divide the data based on the app name, but we restrict these tasks to only those that require interaction with the installed app (e.g., performing a 'forgot password' request). Each domain, along with all associated episodes, is randomly assigned to a train/validation/test split (80/10/10%). We exclude SINGLE, as there are no distinguishable domains across tasks, and GOOGLEAPPS, due to the limited number of distinct apps.

## 5 Experiments

In this section, we report results of two device-control agent models evaluated on AITW. Both models take as input a task instruction, the current screen's pixel-derived features (included in the dataset), and (optionally) a stacked history of screen observations and actions.

### 5.1 Models

**BC.** We implement a Transformer-based [41] Behavioural Cloning (BC) agent. The agent's output is in line with the AITW's data format. It outputs an action type and a gesture. The action type can be *dual-point gesture*, *type*, *go_back*, *go_home*, *enter*, *task_complete*, or *task_impossible*. The gesture

action includes two spatial points, a touch and a lift position. This approach gives this agent a large and flexible action space, as it is able to predict taps and scrolls at arbitrary locations, rather than at specific UI elements as in existing work [4, 28]. We consider two variants of the agent, depending on whether it takes as input the screen-action history (2 prior steps), **BC-history**, or not, **BC-single**. Appendix B.1 provides more implementation details.

**LLM.** We feed to PaLM 2 [3] a textual description of the screen and ask it to predict an action among the supported actions in AITW. We adopt a previously-proposed LLM-based design for device control [43], where the input screen (represented by an Android VH) is converted to HTML syntax. We use a modified version of their prompt (see Appendix B.2), and convert the OCR and detected icons to HTML. We create a zero-shot (**LLM-0**) and a 5-shot Chain-of-Thought (CoT) [44] (**LLM-hist-5-CoT**) version, which also contains history on prior actions taken by the agent, as we observed improves model performance. This model takes the same inputs as the BC model, but as in the original implementation [43], it can only click on detected UI elements, rather than at arbitrary locations and scrolling at precise locations. Since AITW was collected by humans performing precise gestures, some of the recorded gestures are not associated with OCR/Icon-detected UI elements, thus being not feasible for the LLM-based model. This could potentially be ameliorated in future versions by outputting a <x,y> output, rather than tapping specific elements.

## 5.2 Evaluation methodology and metrics

Online evaluation of device-control systems is hard because the execution environment generally does not provide a reward signal. Human validation of such systems can be leveraged, however watching and judging an agent's behaviour in real-time requires constant attention and is error prone. We propose an offline evaluation method which is cheaper and reproducible at the expense of accuracy.

We devise and release the code for *action matching* to evaluate an agent's action's alignment with the ground truth. Two actions can match if their action types are equal. For dual-point taps, they are considered equal if they fall within a 14% screen distance from each other. Alternatively, if the tap actions occur within the same detected bounding box (augmented to 240% of their total size during action matching) they are considered equal. Finally, two dual-point scrolls are equal if they have the same primary scroll axis (vertical or horizontal).

Using action matching, we compute *partial* and *complete* action matching scores (originally proposed by Li et al. [28]). A partial score is defined as the number of correct actions divided by the episode length, and the complete score is defined as a partial match of 1.0.

To validate offline evaluation results, for subsets of the data, we also perform online evaluation. A human marks an episode as failed if any of the agent actions are incorrect, and correct when the agent performs a correct action on every step and achieves the expected goal. Human validation scores typically outperform complete action matching scores due to the multiple valid action alternatives one can take to complete a task. For instance, pressing the navigation bar's back button is functionally similar to using an app-specific back button. As action matching relies on distance-based measures, these actions are deemed distinct.

## 5.3 Results

We evaluate the four agents on the five AITW splits described in §4. For the BC agent, we train and test using all the data. For the LLM agent, due to the high computational overhead, we test on a random sample of 288 episodes for each split. Table 4 reports the average partial matching scores.

The BC agent performs the best across all splits. It performs reasonably well on the OOD tasks, particularly on the subject and verb template splits, indicating the model is generalizing to unseen language instructions and tasks. The LLM-based model only sees a small amount (only those k-shot that are in the prompt) of the training distribution for the OOD experiments. Making use of fine-tuning for future experiments would allow us to leverage more of the training data.

The performance of the LLM-based models suffers due to its element-based action space. For the standard test set, for example, 33% of the episodes have some non-element tap actions (i.e., only <x,y> location), which are infeasible for this modelling approach. Across the feasible actions, LLM-hist-5-CoT has a partial match score of 58%.

| Model | Standard | Out-of-domain generalization | | | |
|---|---|---|---|---|---|
| | | Version | Subject | Verb | Domain |
| BC-single | 68.7 | 59.2 | 64.2 | 66.4 | 52.2 |
| BC-history | **73.1** | **63.2** | **68.5** | **70.4** | **59.7** |
| LLM-0 [43] | 30.9 [25.6, 36.6] | 31.6 [26.3, 37.3] | 33.7 [28.2, 39.5] | 32.6 [27.3, 38.4] | 25.3 [20.4, 30.8] |
| LLM-hist-5-CoT | 39.6 [33.9, 45.5] | 29.5 [24.3, 35.1] | 44.4 [38.6, 50.4] | 41.7 [35.9, 47.6] | 35.8 [30.2, 41.6] |

Table 4: Partial match scores across standard and OOD generalization splits. For the LLM agent, the estimated score and binomial proportion 95% confidence interval are shown. BC evaluation is on the entire test sets; confidence intervals are < 0.1% and are excluded for brevity.

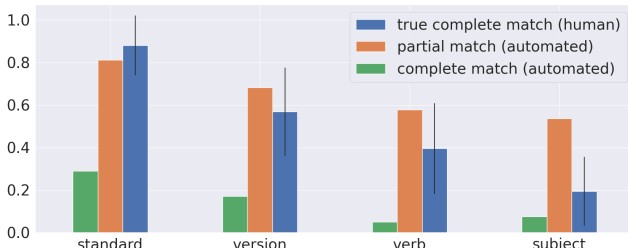

Figure 3: True complete match (estimated using human evaluation), and partial and complete match (both estimated using automated evaluation) for BC-history. True complete match is based on a subset of episodes; 95% confidence bounds are reported. Partial match is correlated with true complete match, while the complete match heuristic is a lower bound score.

We perform human evaluation for BC-history on a small subset from GOOGLEAPPS (on average 86.5 episodes from each split). We use this dataset portion because it has the largest training set, but we exclude the domain split due to the limited number of apps. As shown in Figure 3, we find that action matching is a reasonable approximation of true success rates.

As expected, the agent performs the best on the standard test split. Compared to what is observed across all dataset portions (Table 4) its performance on the standard set is higher, but on subject and verb OOD splits is lower. This is due to the nature of GOOGLEAPPS data (see Table 2) where the tasks are rather distinct (few unique prompts) which makes the verb and subject generalization hard, but at the same time every prompt has many demonstrations, which makes the standard test easier.

Although the automated complete match is low, we note the agent is correct for the majority of steps as indicated by the partial match scores > 0.5. We confirmed this was the case by visual inspection. The agent typically performs many of the initial steps correct, but it is more error prone farther in the trajectory.

In summary, across the four splits, partial match tends to be correlated with true complete match. It is a reliable approximation especially if the number of steps in a task is small. Automated complete metrics represent a lower bound score of the true value.

## 6 Discussion

### 6.1 Data Limitations

**User Demographics Distribution.** The raters are not a representative sample of the entire world population. The screens they visit, containing dynamic content from the Internet, are not representative of the rich variety of content and languages of the world. Similarly, the dataset prompts are exclusively in English, although they could potentially be translated and evaluated using multilingual models.

**Rater device interaction.** Raters use a mouse and keyboard rather than the native touch-based interface. This may result in somewhat different user patterns.

**Form factor.** The dataset is derived from mobile phone user interactions. The dataset could be augmented with more form factors, such as tablets, to increase generalization.

**UI Drift/Evolution.** Our dataset includes an unseen domain split, containing new and unseen UIs, but it may not fully represent the continuous evolution of a given app or website's UI. This dynamic change is an essential aspect of real-world interfaces but is a complex phenomenon to capture comprehensively. However, we do capture some of this drift through the unseen Android version split, reflecting changes in Google apps' UI over various Android versions.

## 6.2 Ethical considerations

**Privacy.** The raters were instructed not to enter any Personal Identifiable Information (PII) during collection. The dataset does not contain any interactions from real users.

**Malicious use.** Malicious actors could use the dataset for undesired purposes such as overriding anti-fraud mechanisms like CAPTCHAs. Malicious actors could also manipulate prompts and/or screen representations of deployed models to achieve undesirable goals.

## 7 Future Work

**Multimodal modeling.** The LLM-based model, adapted from prior work [43], is not as performant as the bespoke BC model. This model consumes a text-based screen representation and cannot output a <x,y> coordinate-based output. A multimodal foundation model [2, 12] that consumes raw pixels and outputs gestures at arbitrary points would be a natural next model type to investigate. Furthermore, any foundation models may benefit from fine-tuning on the AITW training sets.

**Multiple ways to achieve a task.** There are often multiple ways to achieve a task. Future evaluation methods could be more "lenient" and not penalize correct agent actions that do not match human demonstrations. Furthermore, constraining agents to achieve goals in "optimal" ways, however that may be defined, may increase user satisfaction with trained models.

## 8 Conclusions

Mobile device control via natural language commands has broad application. It requires translating high-level instructions into execution plans that operate the device interface as a human would. Recent advancements in general-purpose large foundation models have opened doors for creating such device-control systems, however there remains a substantial void due to the dearth of large, comprehensive datasets essential for training and evaluating these systems.

Addressing these gaps, we present AITW, which is significantly larger and more diverse than existing device-control datasets. AITW consists of 715k episodes across more than 350 Android applications and websites, and a variety of task instructions and execution paths, a realistic representation of real-world system interactions.

Through dataset structure, we provide experimental setups for evaluation under varying conditions, including novel tasks and language, Android versions, and applications and websites. We trained and ran models on the data and demonstrated how to evaluate model performance under novel conditions. We hope AITW will spur research to create more powerful device automation models.

## Acknowledgements

The authors thank Gabriel Taubman, James Stout, Gregory Wayne, and Max Lin for insightful discussions throughout. Thanks to Elisabeth Chauncey for help with dataset release. Thank you to JD Chen for helpful feedback on early manuscript versions. Daniel Toyama, Philippe Hamel, and Anita Gergely provided essential Android environment assistance. We also thank our raters for collecting our data.

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

# Appendix A    Dataset collection

## A.1    Crowdsourcing

This work was carried out by participants who are paid contractors. Those contractors received a standard contracted wage, which complies with living wage laws in their country of employment. Due to global privacy concerns, we cannot include more details about our participants, e.g., estimated hourly wage or total amount spent on compensation.

We provided raters with a detailed instructional document and a video tutorial, followed by having them perform test demonstrations using our system. For multi-step task trajectories, we ensured quality and diversity through manual inspection of a subset of demonstrations. Tasks were marked as complete with the `task_complete` action once a rater completed an assignment, including cases where the task was already completed. In contrast, the `task_impossible` action was used to indicate infeasible tasks, such as turning on a flashlight in an emulator.

For hindsight-language relabeling, we conducted manual reviews of a sample of labeled trajectories due to the nuanced nature of natural language use. The aim was to encourage the creation of descriptive, unambiguous labels and discourage the use of oversimplified technical terms or vague language in order to collect clear and useful task descriptions that cannot be captured using automatic heuristics.

## A.2    Prompt generation

We use the following prompt to extract the subject templates and a similar prompt for the verb templates:

```
# Identify subject variables in commands.
phrase = ["show me popular videos on youtube",
"whats the weather?",
"go to espn.com",
"click the top result",
"open calendar and show me the fourth week of next month",
<INPUT_INSTRUCTIONS>]
result = ["show me {subject1} on {subject2}",
"whats {subject1}?",
"go to {subject1}",
"click {subject1}",
"open {subject1} and show me {subject2} of {subject3}",
```

## A.3    Examples

Example of episodes from AITW are show in Figures 4, 5, and 6.

# Appendix B    Experiment details

## B.1    Behavioral Cloning

The Behavioral Cloning (BC), shown in Figure 7, is a Transformer-based architecture [41] that takes a task instruction, the current screen, and a stacked history of screen observations and actions as input. The model is conditioned on BERT [15] embeddings of the natural language instruction. For the screen input the model embeds each detected OCR and detected icon to a vector of size 512 using the following procedure. We embed the text using a pre-trained BERT model taking the output from the CLS token, which is then linearly projected from size 732 to 512. For the icons, we learn an embedding from the ID, which we add element-wise to the BERT embedding. Following similar approaches [28, 16], we add to this the spatial information by learning four embeddings for each of the bounding box points, which are binned into 200 and 96 elements vertically and horizontally. For the screen history (excluding the current screen), we embed the <x,y> positions of the touch and lift actions, which are added to the element encoding, using a dummy value for non-gesture actions. We found that including action history improves performance.

turn on bluetooth scan

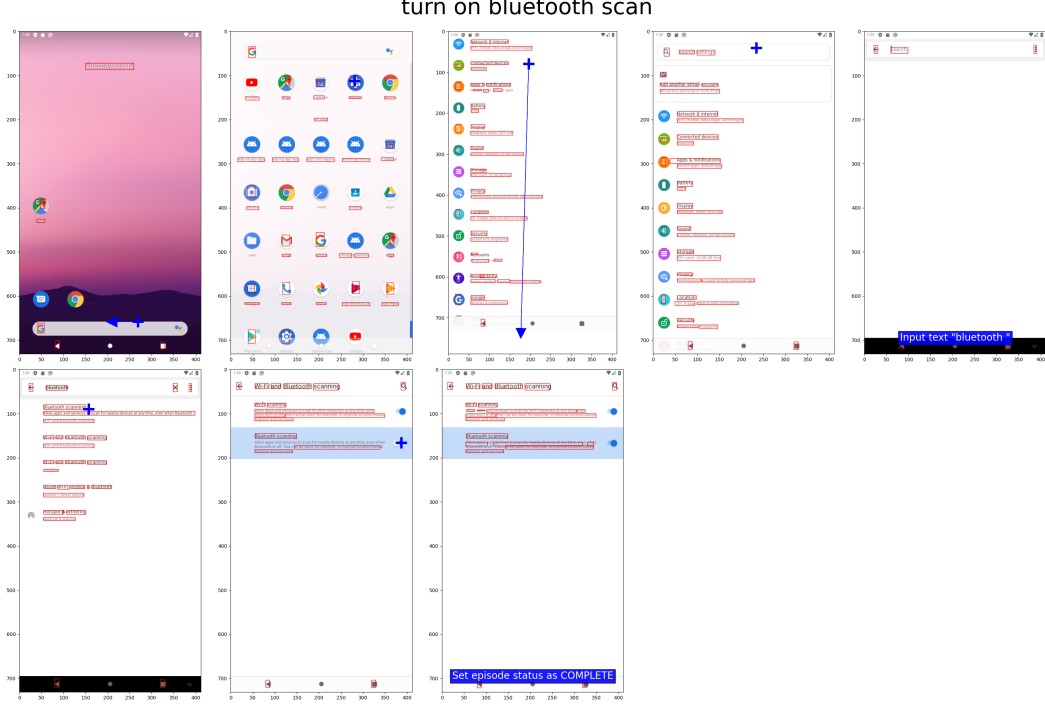

Figure 4: Example episode from the dataset.

What's the news in Argentina?

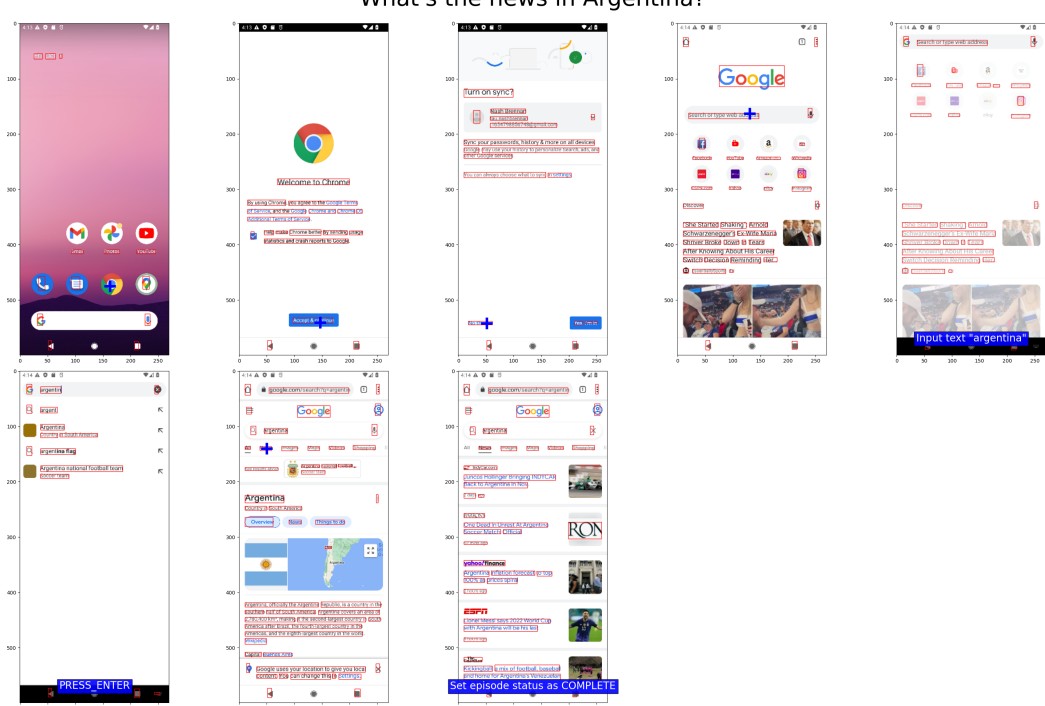

Figure 5: Example episode from the dataset.

clear history in the chrome app

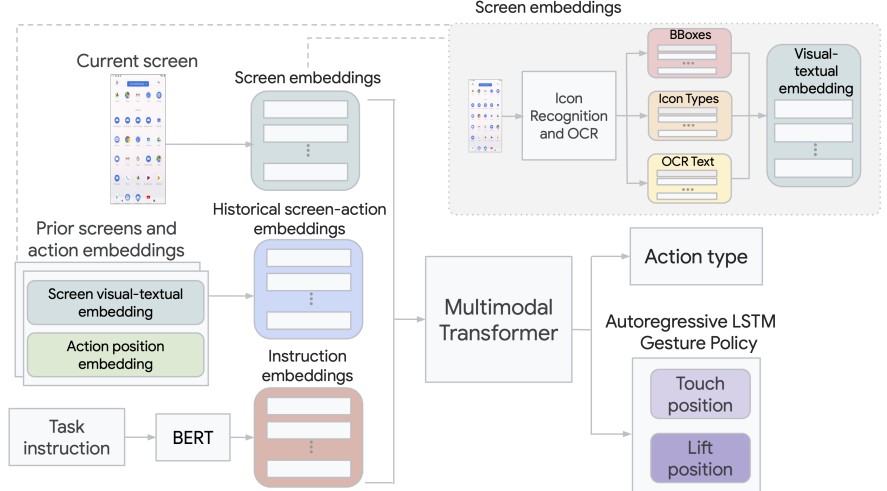

Figure 6: Example episode from the dataset.

Figure 7: Architecture diagram of the BC agent.

Table 5: Partial match scores across generalization splits and datasets for BC-history.

| Dataset | Standard | Version | Subject | Verb | Domain |
|---|---|---|---|---|---|
| GOOGLEAPPS | 75.7 | 63.4 | 48.4 | 57.4 | n/a |
| GENERAL | 63.7 | 52.5 | 65.4 | 64.1 | n/a |
| WEBSHOPPING | 68.5 | 56.7 | 69.5 | 68.4 | 49.6 |
| INSTALL | 77.5 | 66.5 | 76.4 | 77.7 | 66.9 |
| SINGLE | 80.3 | 77.0 | 82.8 | 84.6 | 62.6 |

For gesture actions, the agent outputs a dual-point output. We found such a formulation useful for interacting with many common widgets (e.g., carousel widgets, switching months in a calendar, controlling sliders), which require precise scrolling.

We train the agent using the standard cross-entropy loss using a 2x2 slice of a V2 Tensor Processing Unit (TPU). The agent is implemented using Acme [23], Haiku [22], and JAX [7]. The Transformer has 4 layers, a dropout rate of 0.1, and we train use the AdamW optimizer with a learning rate of 0.0001, and a batch size of 128.

For evaluation we train and perform a hyperparameter search via grid search on the validation set. We choose the best performing model and run it on the test set for the final numbers.

Table 5 reports a breakdown of the performance of the BC-history agent (our best performing agent) across the different dataset splits and portions.

## B.2 LLM

We use the following prompt for LLM-0:

```
Given a mobile screen and a question, provide the action based on the screen
information.

Available Actions:
{"action_type": "click", "idx": <element_idx>}
{"action_type": "type", "text": <text>}
{"action_type": "navigate_home"}
{"action_type": "navigate_back"}
{"action_type": "scroll", "direction": "up"}
{"action_type": "scroll", "direction": "down"}
{"action_type": "scroll", "direction": "left"}
{"action_type": "scroll", "direction": "right"}

Screen:
<SCREEN_REPRESENTATION>
Instruction: <GROUNDING_GOAL>
Answer:
```

We use the following prompt for LLM-hist-5-CoT:

```
Given a mobile screen and a question, provide the action based on the screen
information.

Available Actions:
{"action_type": "click", "idx": <element_idx>}
{"action_type": "type", "text": <text>}
{"action_type": "navigate_home"}
{"action_type": "navigate_back"}
{"action_type": "scroll", "direction": "up"}
{"action_type": "scroll", "direction": "down"}
{"action_type": "scroll", "direction": "left"}
{"action_type": "scroll", "direction": "right"}
```

Previous Actions:
{"step_idx": 0, "action_description": "press [HOME key]"}
{"step_idx": 2, "action_description": "click [Google Icon]"}
{"step_idx": 3, "action_description": "click [search for hotels]"}

Screen:
 </img>
 </img>
<p id=2 class="text" alt="search for hotels"> search for hotels </p>
<p id=3 class="text" alt="in"> in </p>
<p id=4 class="text" alt="mexico city mexico"> mexico city mexico </p>
 </img>
<p id=6 class="text" alt="Share"> Share </p>
<p id=7 class="text" alt="Select alI"> Select alI </p>
<p id=8 class="text" alt="Cut"> Cut </p>
<p id=9 class="text" alt="Copy"> Copy </p>
<p id=10 class="text" alt="hotel in mex"> hotel in mex </p>
 </img>
<p id=12 class="text" alt="best hotel"> best hotel </p>
<p id=13 class="text" alt="mexico city"> mexico city </p>
<p id=14 class="text" alt="in"> in </p>
 </img>
<p id=16 class="text" alt="K"> K </p>
<p id=17 class="text" alt="hotel ciudad"> hotel ciudad </p>
<p id=18 class="text" alt="de mexico"> de mexico </p>
<p id=19 class="text" alt="gran"> gran </p>
 </img>
 </img>
 </img>

Instruction: What time is it in Berlin?
Answer: Let's think step by step. I see unrelated search results in the Google app,
I must clear the search bar, so the action is {"action_type": "click", "idx": 1}

Previous Actions:
{"step_idx": 0, "action_description": "click [DISMISS]"}

Screen:
<p id=0 class="text" alt="Update your"> Update your </p>
<p id=1 class="text" alt="Gmail app"> Gmail app </p>
<p id=2 class="text" alt="attach files from"> attach files from </p>
<p id=3 class="text" alt="To"> To </p>
<p id=4 class="text" alt="download the"> download the </p>
<p id=5 class="text" alt="Drive,"> Drive, </p>
<p id=6 class="text" alt="latest"> latest </p>
<p id=7 class="text" alt="version"> version </p>
<p id=8 class="text" alt="of"> of </p>
<p id=9 class="text" alt="Gmail"> Gmail </p>
<p id=10 class="text" alt="UPDATE"> UPDATE </p>
<p id=11 class="text" alt="DISMISS"> DISMISS </p>
<p id=12 class="text" alt="Got"> Got </p>
<p id=13 class="text" alt="it"> it </p>
 </img>

Instruction: see creations saved in the google photos
Answer: Let's think step by step. I see a popup, I need to open Google Photos, so
the action is {"action_type": "click", "idx": 11}

Previous Actions:

Screen:
<p id=0 class="text" alt="M"> M </p>
<p id=1 class="text" alt="New in Gmail"> New in Gmail </p>
<p id=2 class="text" alt="All the features you"> All the features you </p>

```
<p id=3 class="text" alt="love with"> love with </p>
<p id=4 class="text" alt="a fresh"> a fresh </p>
<p id=5 class="text" alt="look"> look </p>
<p id=6 class="text" alt="new"> new </p>
<p id=7 class="text" alt="GOT IT"> GOT IT </p>

Instruction: open app "Google Play services"
Answer: Let's think step by step. I see the GMail app, I need to open the app
drawer, so the action is {"action_type": "navigate_home"}

Previous Actions:

Screen:
<p id=0 class="text" alt="Tuesday, Aug"> Tuesday, Aug </p>
<p id=1 class="text" alt="9"> 9 </p>
 </img>
 </img>

Instruction: open app "Messenger Lite" (install if not already installed)
Answer: Let's think step by step. I see the home screen, I need to open the app
drawer, I should swipe up, so the action is {"action_type": "scroll", "direction":
"down"}

Previous Actions:
{"step_idx": 0, "action_description": "scroll down"}

Screen:
 </img>
<p id=1 class="text" alt="Search your phone and more"> Search your phone and more </p>
<p id=2 class="text" alt="M"> M </p>
<p id=3 class="text" alt="O"> O </p>
 </img>
<p id=5 class="text" alt="Clock"> Clock </p>
<p id=6 class="text" alt="YouTube"> YouTube </p>
<p id=7 class="text" alt="Photos"> Photos </p>
<p id=8 class="text" alt="Gmail"> Gmail </p>
<p id=9 class="text" alt="All apps"> All apps </p>
<p id=10 class="text" alt="g"> g </p>
<p id=11 class="text" alt="O"> O </p>
 </img>
<p id=13 class="text" alt="10"> 10 </p>
<p id=14 class="text" alt="Calendar"> Calendar </p>
<p id=15 class="text" alt="Camera"> Camera </p>
<p id=16 class="text" alt="Chrome"> Chrome </p>
<p id=17 class="text" alt="Clock"> Clock </p>
<p id=18 class="text" alt="O"> O </p>
<p id=19 class="text" alt="M"> M </p>
<p id=20 class="text" alt="B"> B </p>
 </img>
<p id=22 class="text" alt="Gmail"> Gmail </p>
<p id=23 class="text" alt="Drive"> Drive </p>
<p id=24 class="text" alt="Files"> Files </p>
<p id=25 class="text" alt="Contacts"> Contacts </p>
<p id=26 class="text" alt="G OO"> G OO </p>
 </img>
 </img>
 </img>
 </img>
<p id=31 class="text" alt="Google"> Google </p>
<p id=32 class="text" alt="Maps"> Maps </p>

Instruction: Search for hotels in Chicago.
Answer: Let's think step by step. I see the app drawer, I need to search, so the
action is {"action_type": "click", "idx": 27}
```

```
Previous Actions:
<HISTORY>
Screen:
<SCREEN_REPRESENTATION>
Instruction: <GROUNDING_GOAL>
Answer: Let's think step by step. I see
```

## Appendix C   Dataset format

Each datapoint is stored as a TFRecord file with compression type 'GZIP' with the following fields:

- `android_api_level`: the Android API level of the emulator the episode was collected from
- `current_activity`: the name of the activity running when the example was collected
- `device_type`: the device type of the emulator the episode was collected from, mostly Pixel devices with one custom device image
- `episode_id`: the unique identifier for the episode the example is from
- `episode_length`: the overall number of steps in the episode
- `goal_info`: the natural language instruction the episode is demonstrating
- `image/channels`, `image/height`, `image/width`: the number of channels, height, and width of the screenshot
- `image/encoded`: the encoded screenshot
- `image/ui_annotations_positions`: a flattened array of coordinates representing the bounding boxes of the UI annotations; the coordinates are in (y, x, height, width) format and the length of this array is `4 * num_elements`
- `image/ui_annotations_text`: the OCR-detected text associated with the UI element
- `image/ui_annotations_ui_types`: the type of UI element for each annotation, can be an icon or just text
- `results/action_type`: the type of the predicted action (see 'Action space' for more details)
- `results/type_action`: if the action is a `type` then this holds the text string that was typed
- `results/yx_touch`, `results/yx_lift`: the (y, x) coordinates for the touch and lift point of a dual point action
- `step_id`: the example's zero-indexed step number within the episode (i.e. if `step_id` is 2, then this is the third step of the episode)

