# OpenReview forum: "AndroidInTheWild: A Large-Scale Dataset For Android Device Control"
_NeurIPS.cc/2023/Track/Datasets_and_Benchmarks — NeurIPS 2023 Datasets and Benchmarks Poster_

### Official Review · Reviewer_2zAP · 2023-07-18
**A larger dataset mobile device control**

**Rating:** 8
**Confidence:** 4
**Clarity:** The paper is well-written and easy to…

**Strengths:**

This dataset is at least an order of magnitude larger than existing datasets. This will provide other researchers to train, and test device automation models. This is very timely given the popularity of using Large Language Models to automate tasks.

**Additional Feedback:**

Thanks for the response. I remain positive for accepting this paper.

**Correctness:**

The data collection process is clearly outlined in Figure 1 and explained well in sections 3.1 and 3.2 & 3.3.
Two baseline models were developed and evaluated with the proposed dataset.
Overall, I am convinced that this would be a good dataset for other researchers.

**Documentation:**

The dataset is released under Creative Commons license. The data set is well documented.

**Ethics:**

There are no major ethical concerns. It is good to understand if these data collection studies have to go through an ethical clearance process within Google Research. Including a statement on that in the data collection process would be good.

**Limitations:**

The authors have not indicated any limitations. At least, the potential bias in the dataset could be mentioned as a limitation.

**Opportunities For Improvement:**

Authors could provide more details of the distribution of the users demographics in the data collection process so that the reader can understand if the dataset has any potential bias from a specific user group.

**Relation To Prior Work:**

Table 1 and section 2.1 provides a clear comparison of this dataset to existing datasets.

**Summary And Contributions:**

The authors present a large dataset that can be used to develop models to train and evaluate automation models for mobile device control.

The authors contribute with a large dataset, a detailed analysis of the generalization capability of the dataset, and the performance of two baseline models

---

> ### Author Response · Authors · 2023-08-21
> **Thank you reviewer 3**
>
> Thanks a lot for your careful review and valuable suggestions. We updated the paper by and reply to the concerns one by one. Specifically, we've added Sections 6 Discussion to address the comments.
>
> *Missing Limitations; User Demographics Distribution & Potential Bias*: Your point regarding potential biases in the dataset is well taken. We have added a Data Limitations section and explicitly discussed user demographics.
>
> *Ethical Clearance Process within Google Research*: We recognize the importance of ethical considerations in our research and your suggestion regarding ethical clearance is appreciated. We have added a discussion subsection on privacy to reflect this. Our dataset went through an approval process. However, details about specific internal processes often fall under proprietary information and may not be appropriate for inclusion in a public document.

---

### Official Review · Reviewer_t95d · 2023-07-20
**Nice work! Useful dataset!**

**Rating:** 7
**Confidence:** 4
**Correctness:** Correct
**Clarity:** Clear

**Strengths:**

1. Comprehensive dataset: The AndroidInTheWild dataset is large and varied, covering multiple Android versions and device types. This makes it a valuable resource for researchers and developers working on mobile device control systems.
2. Baseline models: The paper provides two baseline models using Behavioural Cloning and LLMs, which can be used to evaluate the generalization capabilities of device automation models. This provides a useful starting point for researchers and developers looking to improve their own models.


**Additional Feedback:**

See strength and weakness.

**Documentation:**

Yes, there is sufficient detail on data collection and organization, availability and maintenance.

**Ethics:**

No concern.

**Limitations:**

1. The main limitation of this baseline model is that it can only interact with detected UI elements, rather than arbitrary locations, and it relies on OCR/Icon detection, making certain gestures infeasible. To overcome these issues, future versions could be improved by outputting a <x,y> coordinate-based output, providing more flexibility for interaction with any location on the screen.
2. The paper discusses four experimental setups to evaluate the generalization of agents to new scenarios. However, there are limitations in the evaluation, such as small test splits, incomplete assessment of unseen scenarios, potential language pattern omissions, and possible dataset biases.


**Opportunities For Improvement:**

The main limitation of this baseline model is that it can only interact with detected UI elements, rather than arbitrary locations, and it relies on OCR/Icon detection, making certain gestures infeasible. To overcome these issues, future versions could be improved by outputting a <x,y> coordinate-based output, providing more flexibility for interaction with any location on the screen.

**Relation To Prior Work:**

ANDROIDINTHEWILD significantly improves upon these prior datasets in various aspects. It is two orders of magnitude larger than MoTIF, containing 715,142 episodes and spanning 30,378 unique prompts. The dataset covers a wide range of tasks involving Android apps and websites, four versions of Android, and eight different device types. Furthermore, it introduces random starting states for some demonstrations, which were not provided in previous datasets. Observations in ANDROIDINTHEWILD consist of screenshots and pixel-based screen features, making it a more comprehensive and diverse dataset for device control research.

**Summary And Contributions:**

The paper describes a new dataset called AndroidInTheWild, which is designed to help researchers and developers improve mobile device control systems. The dataset includes over 700k episodes and 30k unique prompts, and covers a wide range of Android versions and device types. The authors also provide two baseline models using Behavioural Cloning and LLMs, which can be used to evaluate the generalization capabilities of device automation models. The paper concludes with a checklist for authors to ensure that their work accurately reflects the contributions and scope of the paper, and discusses potential limitations and future work.

---

> ### Author Response · Authors · 2023-08-21
> **Thank you reviewer 2**
>
> Thanks a lot for your careful review and valuable suggestions. We updated the paper and reply to the concerns one by one. Specifically, we've added Sections 6 and 7, Discussion and Future Work, to address the comments.
>
> *Limitations of Baseline Model*: Your suggestion regarding the improvement of the baseline model by outputting <x,y> coordinate-based output was insightful. We've discussed this idea in the "Research Topics" section and look forward to exploring it in future work.
>
> *Experimental Setups and Evaluation Limitations*: We appreciate your insights regarding the evaluation splits' limitations, and we agree with some of your points. However, it's crucial to emphasize that our primary focus in this paper is on presenting the dataset, demonstrating its potential, and conducting an initial generalizability analysis. We acknowledge that the evaluation could be further refined, and the limitations you've pointed out represent important considerations for future work. We envision subsequent research building upon our foundational effort, elaborating on more nuanced evaluation techniques, and exploring the creation of new or larger data splits. We believe that the limitations do not detract from the value and novelty of our dataset but rather open exciting avenues for continued exploration.

---

> > ### Comment · Reviewer_t95d · 2023-08-29
> >
> > Thanks the author's for their effort on rebuttal. The authors address my concern and I decide to keep my rating.

---

### Official Review · Reviewer_YSqi · 2023-07-21
**Review for Android In The Wild paper**

**Rating:** 8
**Confidence:** 4
**Correctness:** The claims seem to be well supported …

**Strengths:**

* The paper is really well organised, has clear motivation and goals and provides a nicely structured overview of related work and methodology.
* The amount of episodes, annotations and diversity of devices and OS versions are remarkable.
* I particularly liked the complexity of the tasks compared to previous approaches with respect to description of actions not tied to UI elements.

**Additional Feedback:**

* There are certain actions that can be accomplished in multiple ways. Is there a notion of optimality that could be integrated in the learning process?
* Would it be possible to have cross-language functionality, e.g. Natural Language in English and interface in German or vice-versa?
* How would such an automation feature integrate with something like Apple Shortcuts (or equivalent in Android)?

**Clarity:**

The paper is very well written, nicely structured and easy to follow. No issues whatsoever.

**Documentation:**

Both the repository documentation and the appendix of the paper provide good documentation on the methodology and the dataset creation. One missing thing from the codebase is the lack of code for the results in §5.3.

**Limitations:**

The limitations of this work have been dispersed throughout the document. However, there are certain missing aspects; in particular wrt:
* Privacy (if model resides at the cloud)
* Size of model to reside locally on-device upon deployment (if model resides locally)
* Behaviour of the model under different application and website versions (More dynamic that OS version changes).

Wrt negative societal impact, the authors have not provided such positioning. Although the dataset might not directly have societal impact, it can act as an enabler for malicious actors to perform undesired actions, by manipulating the prompt or the screen elements (e.g. in high frequency). Moreover, another potential downside is the application of such techniques to solve tasks shown in CAPTCHAs and thus overriding anti-fraud mechanisms. Last, since LLMs (e.g. Palm-2) are particularly large in size, they would probably reside in the cloud. Thus, there are privacy extensions that have not been mentioned in the manuscript.


**Opportunities For Improvement:**

* The performance of the PALM-2 baseline is not doing very well, particularly wrt OOD generalisation.
* Human annotators are interacting with the Android phones through a mouse and a pointer, which is very different to the native interaction. I am not sure what side-effect this can cause to the level of interaction.
* It would be interesting to have different form factors of devices integrated, such as a pixel tablet to see generalisation to even more diverse setups.
* It would be further interesting to see an extension where a multimodal LLM takes the raw screenshot as input and acts upon it, instead of a hierarchical representation of the current screen.

**Relation To Prior Work:**

The background and related work are well organised and provide good prior work coverage. If anything, it would be nice to explictly refer to examples of work accessing APIs through NLP, such as [1,2]

[1] Schick, T., Dwivedi-Yu, J., Dessì, R., Raileanu, R., Lomeli, M., Zettlemoyer, L., ... & Scialom, T. (2023). Toolformer: Language models can teach themselves to use tools. arXiv preprint arXiv:2302.04761.
[2] Patil, S. G., Zhang, T., Wang, X., & Gonzalez, J. E. (2023). Gorilla: Large language model connected with massive apis. arXiv preprint arXiv:2305.15334.

**Summary And Contributions:**

The current paper tackles the problem of automating the control of mobile devices given a task expressed in natural language. To this direction, the authors create a dataset of 30k prompts, sourced by humans, LLMs and technical documentation, yielding 715k episodes across four Android versions and eight different Pixel devices.

Contrary to previous datasets, the action space involves precise gestures not necessarily included as-is in the prompt. Lastly, they provide two baseline transformer-based agents and report performance on task fulfilment.

---

> ### Author Response · Authors · 2023-08-21
> **Thank you reviewer 1**
>
> Thanks a lot for your careful review and valuable suggestions. We updated the paper and reply to the concerns one by one. Specifically, we've added Sections 6 and 7, Discussion and Future Work, to address the comments.
>
> *Performance of PALM-2, OOD Generalization, and Multi-modal LLMs*: We appreciate your feedback on this aspect, and we acknowledge that this is an area for future research and enhancement. We have highlighted in the new future work section ways to improve the model using an arbitrary <x,y> action space and multimodal models. Given that the focus of this paper is on the dataset and not necessarily the optimization of specific models, we feel this is a suitable way to address this issue. Given our baseline was an implementation of an existing model, we think focusing on that model will help readers put the results in context.
>
> *Rater Device Interaction & Form Factor*: We've added a subsection in the discussion to acknowledge these limitations. Your feedback on this has helped us emphasize these considerations.
>
> *Privacy and Malicious Use*: We included a section on privacy and ethics considerations to address your concerns about potential negative societal impacts, privacy, and malicious use of the data.
>
> *Code for results in §5.3* We acknowledge the value in releasing this part of the codebase, and we appreciate your interest in it. Unfortunately, due to certain internal constraints, we are unable to make this particular portion available at this time.
>
> *Cross-Language Functionality:* As you suggested, we found this to be an interesting idea and have added a point mentioning that the prompts are in English. We highlight that future work could involve translation and using Multilingual models.
>
> *Privacy and model size concerns:* Thank you for highlighting the concerns regarding privacy and model size. While we recognize the importance of these aspects, particularly in the context of on-device deployment, the primary focus of this paper remains on the dataset itself. The considerations of privacy have been addressed within the relevant section, and the topic of deployment model size, although relevant, falls outside the scope of our present investigation.
>
> *Behaviour under different application and website versions*: Unfortunately, our dataset does not provide this kind of information. We only have captured demonstrations across different versions of Android. We have acknowledged this limitation in the Data Limitations section we have added.
>
> *Multimodal Modeling & Multiple Ways to Achieve a Task*: We have expanded the discussion on these points, recognizing them as valuable future research directions.
>
> *API use research*: We've added a reference to the work you mentioned on L108. Thank you for raising this point as it's very topical and relevant.

---

> > ### Comment · Reviewer_YSqi · 2023-08-29
> > **Reply to authors**
> >
> > Dear authors,
> >
> > Thank you for the clarifications and for incorporating my feedback in the manuscript. I am happy with the paper, so I am keeping my score as is.

---

### Decision · Program_Chairs · 2023-09-22

**Decision:**

Accept (Poster)

**Comment:**

All the reviewers unanimously have voted to accept the paper. Congrats!